# Evaluation of Polyphenols Synthesized in Mature Seeds of Common Bean (*Phaseolus vulgaris* L.) Advanced Mutant Lines

**DOI:** 10.3390/ijms25115638

**Published:** 2024-05-22

**Authors:** Teodora G. Yaneva, Wieslaw Wiczkowski, Andrey S. Marchev, Dida Iserliyska, Milen I. Georgiev, Nasya B. Tomlekova

**Affiliations:** 1Department of Food Technology, Institute of Food Preservation and Quality, Agricultural Academy, 154 Vasil Aprilov Blvd., 4027 Plovdiv, Bulgaria; 2Department of Chemistry and Biodynamics of Food, Institute of Animal Reproduction and Food Research, Polish Academy of Sciences, 10 Tuwima Str., 10-748 Olsztyn, Poland; 3Laboratory of Metabolomics, Department of Biotechnology, Stephan Angeloff Institute of Microbiology, Bulgarian Academy of Sciences, 139 Ruski Blvd., 4000 Plovdiv, Bulgaria; 4Laboratory of Molecular Biology, Department of Breeding, Maritsa Vegetable Crops Research Institute, Agricultural Academy, 32 Brezovsko Shosse Str., 4003 Plovdiv, Bulgaria

**Keywords:** common bean (*Phaseolus vulgaris* L.), anthocyanins, seed color, HPLC-MS/MS, EMS-induced mutagenesis

## Abstract

This study aimed to investigate the availability of flavonoids, anthocyanins, and phenolic acids in mutant bean seeds, focusing on M_7_ mutant lines, and their corresponding initial and local cultivars. HPLC-DAD-MS/MS and HPLC-MS/MS were used to analyze twenty-eight genotypes of common bean. The obtained results suggest that the mutations resulted in four newly synthesized anthocyanins in the mutant bean seeds, namely, delphinidin 3-*O*-glucoside, cyanidin 3-*O*-glucoside, pelargonidin 3-*O*-glucoside, and petunidin 3-*O*-glucoside, in 20 accessions with colored seed shapes out of the total of 28. Importantly, the initial cultivar with white seeds, as well as the mutant white seeds, did not contain anthocyanins. The mutant lines were classified into groups based on their colors as novel qualitative characteristics. Five phenolic acids were further quantified: ferulic, *p*-coumaric, caffeic, sinapic, and traces of chlorogenic acids. Flavonoids were represented by epicatechin, quercetin, and luteolin, and their concentrations in the mutant genotypes were several-fold superior compared to those of the initial cultivar. All mutant lines exhibited higher concentrations of phenolic acids and flavonoids. These findings contribute to the understanding of the genetics and biochemistry of phenolic accumulation and anthocyanin production in common bean seeds, which is relevant to health benefits and might have implications for common bean breeding programs and food security efforts.

## 1. Introduction

Common bean (*Phaseolus vulgaris* L.) is cultivated globally as a staple crop rich in vital nutrients, such as complex proteins, minerals (iron, zinc, magnesium, and potassium), dietary fiber, and vitamins. It is also low in fat and cholesterol, making it a healthy and nutritious food option [1]. Consumption of common bean has many benefits related to obesity, heart disease and diabetes prevention, and lower risk of certain types of cancer [1,2]. Human health and longevity are generally improved by polyphenol-rich foods, i.e., fruits, vegetables, whole grains, and beans [3]. Consumption of dry beans in Bulgaria has a long tradition in the country’s history. Traditionally, consumer preferences have included mostly white bean seeds, which led to the white-colored coat prevailing among local cultivars and breeding lines [4]. They are high in fiber, protein, folate and other B-vitamins, minerals, and antioxidants [5]. Hence, the diversity of common bean in Bulgaria is narrow, and to overcome these limitations and to create new genetic diversity, breeders are in search of new approaches [6]. Recently, colored common bean seeds have attracted great attention because of their strong health-promoting effects [2,7].

Anthocyanins appear to be responsible for giving common bean its characteristic colors, such as black, purple, blue, orange, and red [2,8]. They are synthesized in the seed coat [9] and play a part in the resistance of common bean to various stresses, including pests, diseases, and environmental stresses such as high temperatures and drought [10]. Anthocyanins are water-soluble pigments commonly found in vegetables, fruits, and flowers, and are responsible for the coloration of their tissues [7,10,11,12]. The bright colors of anthocyanins help the plant attract pollinators, act as sunscreen against ultraviolet (UV) light [11,13], and protect the plants from pests and pathogens due to their antimicrobial and antifungal properties. More than 635 anthocyanins are found in nature, which are glycosylated forms of over 20 known anthocyanidins, of which only 6 are considered to be the dominant pigments in the plant kingdom. These common anthocyanidins include pelargonidin, cyanidin, malvidin, petunidin, peonidin, and delphinidin, which are responsible for the vibrant colors in various plant tissues [14]. To date, five of these prevalent anthocyanidins have been identified in the seed coats of common bean, with the only exception being peonidin [11,15].

Аnthocyanins are glucosides of anthocyanidins (positively charged flavonoids) that are chemically unstable under high temperature, light, and extreme pH values. They are found in more stable glucoside form in nature and differ in hydroxylation and methylation of the hydroxyl (OH) groups of the B-ring at 3′ and 5′ positions. The glycosylation and acylation of the C-ring stabilize the molecule and affect its biological activity [11,14,16]. The color of these flavonoids depends on the pH of the media, being more stable in an acidic environment. Anthocyanins could be used as natural colorants and antioxidants (E 163) in the food and beverage industry [11]. Therefore, their availability would be a valuable quality trait in a new bean genotype, and could be further developed as a new mutant cultivar [2,7,8,10,12,17]. Breeding programs have been conducted to enhance the quality of existing and develop new superior bean genotypes in terms of food and feed [18].

Secondary metabolites, such as anthocyanins, phenolic acids, and flavonoids, are very desirable as a part of the diet; therefore, this work aimed to detect their concentrations (accumulation) in bean seeds from M_7_ mutant lines as a stage of the bean mutation breeding program at the Maritsa Vegetable Crops Research Institute in Plovdiv, Bulgaria. The mutant lines with improved traits were selected for further research. The concentrations of polyphenols in dry bean seeds vary widely, and their profile is influenced by both genotype and environmental conditions. The genotype can be altered by mutagenesis to generate genetic diversity and develop new bean lines and cultivars with improved traits with respect to improved nutritional quality.

As already stated, common bean diversity in Bulgaria is rather narrow and bean production is reduced dramatically [19]. To overcome these limitations, genetic plant breeders have been in search of new approaches for developing promising genotypes [6].

One of the important tools to generate genetic variation for common bean breeding programs is chemical mutagenesis [20,21]. Ethylmethane sulfonate (EMS)-induced mutagenesis in common bean has been successfully used to generate mutants with various altered traits [5]. These traits can encompass plant architecture, flower and seed coat color, seed development, and biological nitrogen fixation, and alterations in phytic acid biosynthesis are studied in genetic screening programs [1,22,23]. EMS-induced mutagenesis is a valuable tool in common bean breeding, allowing the development of diverse mutant populations and the study of genetic changes. This approach can lead to the development of new plant cultivars with desirable traits for agriculture and research purposes. It is essential to conduct a thorough screening and selection to ensure that the desired traits are retained while minimizing any negative effects from the mutagenesis process [21,22].

At an early stage of the present work, EMS-induced mutagenesis was applied to enrich the gene pool of *P. vulgaris* L. by obtaining new genotypes. Until now, in Bulgaria, the concentrations of anthocyanins, phenolic acids, and flavonoids in bean seeds, as well as in mutants (with a changed color), have not been analyzed.

## 2. Results

### 2.1. Profiles of Anthocyanins Acquired by HPLC-DAD-MS/MS

The chromatographic evaluation of the purified bean extract registered at 520 nm showed the presence of three major anthocyanins, namely, delphinidin 3-*O*-glucoside, cyanidin 3-*O*-glucoside, and pelargonidin 3-*O*-glucoside. Traces of petunidin 3-*O*-glucoside were found only in M 2.2 and M 2.3 (with concentrations of 35.79 µg/g dry weight (dw) and 24.26 µg/g dw), while it was present in no other mutant line or cultivar, which makes these two unique. The genotypes were analyzed using a standard solution of malvidin 3-*O*-glucoside at R_t_ of 33.67 min and none of the genotypes examined possessed this anthocyanin.

#### 2.1.1. Chromatographic Analysis

The data from the chromatographic tests of the mutant bean extracts scanned at 520 nm showed the presence of three major anthocyanins in the EMS-induced mutant lines. Their concentrations are presented in Table 1 and Figure 1.

The most abundant anthocyanin detected in the mutant bean lines was delphinidin 3-*O*-glucoside, with the highest concentration found in M 2.3 (1887.18 µg/g dw), followed by M 15 (1507.83 ± 65.74 µg/g DW), M 17 (1158.97 ± 38.7 µg/g dw), M 8.3 (1092.44 µg/g), and M 12 (1023.63 µg/g). This anthocyanin was detected in neither the initial nor the local cultivar (see Appendix A).

The next valuable substance found was cyanidin 3-*O*-glucoside, which had the highest concentration in M 8.3 (618.52 ± 24.43 µg/g dw), followed by M 15 (581.38 ± 14.38 µg/g dw), M 17 (537.84 ± 17.23 µg/g dw), and M 12 (451.75 ± 21.11 µg/g dw). Cyanidin 3-*O*-glucoside concentrations in M 2.3 and in the local cultivar were 60.40 µg/g dw and 76.97 µg/g dw, respectively, and it was absent in the initial cultivar.

In most of the colored mutant lines, pelargonidin 3-*O*-glucoside was present in concentrations from 304.54 ± 3.52 µg/g dw in M 2.2 to 441.41 ± 11.78 µg/g dw in M 8.3, while in M 15 and M 2.3 it was 390.63 ± 81.74 and 312.76 ± 11.52 µg/g dw, respectively. The described result of EMS-induced mutagenesis was demonstrated by the fact that the mutant lines contained anthocyanins that were not present in the initial cultivar. White, yellow, and beige seeds, as well as some light-brown seeds (M 2.1, M 3, M 7.1, M 8.1, M 10, M 11.1, and the initial white seeded cultivar “Evros”), did not contain anthocyanins according to the method described.

The most promising mutant lines concerning the anthocyanin availability were genotypes M 2.3, M 8.3, M 12, M 15, and M 17.

M 15 contained the second highest availability of delphinidin 3-*O*-glucoside and cyanidin 3-*O*-glucoside, compared to all accessions. Pelargonidin 3-*O*-glucoside was available at almost equal levels in most of the genotypes. The M 17 mutant line was also evaluated as a promising genotype, with delphinidin 3-*O*-glucoside, cyanidin 3-*O*-glucoside, and pelargonidin 3-*O*-glucoside concentrations of 1158.97, 537.84, and 449.43 µg/g dw, respectively. In M 12, the concentrations of the anthocyanins were, respectively, 1023.63, 451.75 and 404.23 µg/g dw. M 8.3 contained a lower concentration of delphinidin 3-*O*-glucoside (1092.44 µg/g dw) but was the richest in cyanidin 3-*O*-glucoside (618.52 µg/g dw) and pelargonidin 3-*O*-glucoside (441.41 µg/g dw) among the genotypes.

Although the level of cyanidin 3-*O*-glucoside was lower in M 2.3 than in the other mutant lines, the highest concentration of delphinidin 3-*O*-glucoside and the presence of petunidin 3-*O*-glucoside indicates that the M_8_ line is very promising material for further research and development of mutant cultivars.

#### 2.1.2. MS Spectroscopy

The four peaks identified and quantified by the conducted HPLC analysis were further confirmed to be anthocyanins with MS spectral signals at *m*/*z* 271.1, 287.1, 303.1, and 317, which correspond to the molecular ions of the anthocyanidins pelargonidin, cyanidin, delphinidin, and petunidin, respectively (Table 2).

The mass spectrometry (MS/MS) analysis of the four compounds tentatively identified glucosides of the anthocyanidins as delphinidin, cyanidin, petunidin, and pelargonidin (Appendix A).

Peak 1 was identified as delphinidin-glucoside (myrtillin), as it possessed a molecular ion [M-glu]+ with an *m*/*z* (mass-to-charge ratio) value of 465.2 mass units (mu) and a fragment ion with an *m*/*z* value of 303.1 mu. The loss of 162 mu corresponds to the loss of one molecule of dehydrohexose. Peak 2 was identified as cyanidin-glucoside because its molecular ion [M-glu]+ had an *m*/*z* value of 449.2 mu and a fragment ion [M]+ with an *m*/*z* value of 287.1 mu. Again, the loss of 162 mass units corresponds to the loss of one molecule of dehydrohexose (glucose). Peak 3 was identified as petunidin-glucoside with a molecular ion [M-glu]+ with an *m*/*z* value of 479.2 m/u and a fragment ion [M]+ of 317.1 mu, and the loss of 162.1 mass units corresponds to the loss of one molecule of dehydrohexose. Peak 4 was identified as pelargonidin-glucoside because its molecular ion [M-glu]+ had an *m*/*z* value of 433.2 mu and its fragment ion [M]+ had an *m*/*z* value of 371.1 mu. The same loss of 162.1 mass units corresponds to the loss of one molecule of dehydrohexose. These identifications were based on the observed mass spectra and the characteristic loss of 162 or 162.1 mu corresponding to the removal of a glucose (dehydrohexose) moiety.

### 2.2. Seed Color Evaluation

#### 2.2.1. Cluster Analysis—Seed Clusters Based on Coat Color

Initially, an attempt was made to classify the mutant bean seeds into different color groups using k-means cluster analysis for the selection of the best number of clusters from the data. Consequently, each genotype was assigned to a specific color group designated I to IX. Table 3 shows the genotype and the group to which the analyzed accessions belong, together with the seed color percentage. Nine groups differing by color were identified: dark brown to black (I), white (II), light brown to brown (III), brown to ink (IV), light brown to dark brown (V), pale yellow (VI), beige (VII), speckled beige to light brown (VIII), and speckled brown (IX). Among the twenty-six genotypes investigated, twenty-four were segregated by color and within each group there were small percentages considered to be marginal.

Mutant genotypes M 15 and M 8.3 had the highest number of black seed coats (95.77% and 91.04%, respectively) followed by M 9.2 (75.47%), M 7.3 and M 17 (70%), and M 12 (69.70%) and M 2.3 (56.86%). The dark-brown color was the trait of 16.2., M 6.2, and M 8.1 (67.14%; 58.73%; 52.73%). The mutant lines M 7.2 (51.16%), M 8.2 (72.41%), M 9.1 (50.00%), and M 11.2 (62.79%) were brown and M 6.1 was speckled brown (50%); M 8.2 had the highest number of light-brown seeds; M 11.1 was beige; and M 7.1 was speckled beige. Mutants M 2.1 and M 3 were white and the initial cultivar had a pale-yellow color. The heterogeneous color mixtures were characteristic of M 19.1 and 19.2, as well as M 14. These data are presented in the Appendix A (Appendix A).

#### 2.2.2. Hierarchical Cluster

The cluster diagram of 25 mutant bean (*P. vulgaris* L.) genotypes, and initial and local cultivars, based on the color parameters L and C, is shown in Figure 2. The dendrogram demonstrates the linkage distance amongst all the investigated genotypes grouped in three clusters, comprising a linkage map. Moving from the right to the left point of the x-axis of the tree diagram, the first cluster, considered as Cluster I, contained three genotypes—M 2.1, M 3, and the initial cultivar—which all have white-colored seeds. Cluster II contained eight genotypes—M 2.2, M 6.1, M 7.1, M 10, M 11.1, M 19.1, the local cultivar, and M 8.1, which was the outlier in this group, being colored dark brown. Mutants M 6.1 and M 11.1 with light-brown to beige seeds were at a larger linkage distance compared to the rest in the same cluster. Cluster III comprised 16 mutant bean genotypes—M 2.3, M 6.2, M 7.2, M 7.3, M 8.2, M 8.3, M 9.1, M 9.2, M 11.2, M 11.3, M 12, M 14, M 15, M 16.2, M 17, and M 19.2, whose seed coat colors were represented by all tints of brown through black to ink. Some authors applied a similar approach to evaluate the color of Mexican wild and weedy common bean cultivars by building color groups based on similarity [24].

#### 2.2.3. Mean Cluster Values

The cluster means of 25 mutant bean (*P. vulgaris*) genotypes, and initial and local cultivars, for their color parameters are displayed in Figure 2. Mean values for luminosity were the highest in Cluster I (81.7), much lower in Cluster II (30.11), and the lowest in Cluster III (17.95). The parameter chroma followed the same tendency. As noted in the colorimetric measurements, the grain luminosity (L) for the mutant beans ranged from 79.34 to 15.77 and the chroma (C) from 1.66 to 25.58. The white and pale-yellow seed coats had the highest luminosity and chroma values, whereas the lowest levels were characteristic of the black seed coats (see Appendix A). Luminosity and chrome correlation (0.42, *p* ≤ 0.05) is displayed in the Appendix A.

#### 2.2.4. Analysis of Variance

Analysis of variance for color parameters L and C is shown in Table 4. The selected parameters were found to be significant based on the F values, significance level *p* ≤ 0.1, and *p*-values less than 0.1. For instance, Sammyia et al. reported highly significant results for 36 common bean genotypes’ quantitative attributes [25].

#### 2.2.5. Phenolic Acid and Flavonoid Profile by HPLC-MS/MS Analysis (QTRAP 5500 Ion Trap Mass Spectrometer)

Freeze-dried biomass from the accessions was analyzed by means of HPLC–MS/MS to determine various forms of phenolic acids and flavonoids. The profile and concentration of phenolic acids and flavonoids are presented in Table 5. Phenolic profiling by HPLC revealed epicatechin and quercetin as the most abundant flavonoids. Among phenolic acids, *p*-coumaric and caffeic acids were found in higher concentrations. The levels of phenolic acids and flavonoids varied significantly with the genotype.

The data for the concentrations of phenolic acids and flavonoids in the bean extracts showed the presence of five phenolic acids and three flavonoids in the EMS-induced mutant lines. The most abundant among the phenolic acids was *p*-coumaric acid in the initial cultivar “Evros” (43.23 μg/g dw), and ranging from 145.56 to 1301.31 μg/g dw in M 10 and M 7.1, respectively. The concentration of ferulic acid was 156.58 μg/g dw in “Evros”, which was almost the same as the concentration of 158.87 μg/g dw in M 15, and reached 816.67 μg/g dw in M 6.2 and up to 934.84 μg/g dw in the white mutant line M 3. The concentration of synaptic acid in the initial cultivar was 41.16 μg/g dw, and it was the only phenolic acid with lower concentrations in most of the mutant lines. However, higher concentrations of synaptic acid were quantified in M 2.3, M 6.1, M 7.1, M 8.3, and M 11.2, with the highest in M 3 (respectively, 101.90, 359.54, 417.13, 165.00, 311.67, and 435.55 μg/g dw). Caffeic acid had the lowest concentration of 53.10 μg/g dw in “Evros”, and ranged from 180 to 772 μg/g dw in M 19.1 and M 2.1. Minor amounts of chlorogenic acid, starting from 0.17 μg/g dw in “Evros”, and reaching 2.90, 2.98, 3.10, and 3.29 μg/g dw in M 10, 11.2, M 8.2, and M 9.2, respectively, were identified. M 3, M 6.1, M 7.1, and M 7.3 showed the highest total phenolic acid concentration (from 1916.66 to 2796.94 μg/g dw). The major flavonoid was epicatechin, with a concentration of 8.00 μg/g dw in “Evros”; its concentrations in all the mutant lines were higher than this value, ranging from 10.18 μg/g dw in M 17 to 88.46 μg/g dw and 86.18 μg/g dw in M 1.11 and M 2.3. Quercetin accounted for 4.82 μg/g dw in “Evros” and ranged from 4.15 μg/g dw in M 19.1 to 84.19 μg/g dw in M 15; the other mutant abundant in quercetin was M 14, with 79.78 μg/g dw. Luteolin was present in lower concentrations. It was the lowest in M 19.2 and “Evros” (0.34 μg/g dw and 0.47 μg/g dw), and the highest concentrations were found in M 7.1 and M 7.2, at 9.75 μg/g dw and 9.09 μg/g dw, respectively. The best flavonoid/phenolic acid ratios were observed in M 15, M 11.1, M 14, M 8.2, and M 9.2, of 0.108, 0.102, 0.086, 0.082, and 0.073, respectively, which makes these mutant lines excellent candidates for developing new cultivars.

The evaluation of the best accessions according to their flavonoid/phenolic acid ratio contributes to their selection for the provision of health benefits and will have implications for enhancing food quality by improving common bean breeding programs.

## 3. Discussion

### 3.1. Anthocyanin Concentration in Common Bean

The biosynthetic pathway for anthocyanins is well resolved across plant crop species. Most of the genes are already cloned to introduce new traits into crop cultivars and increase anthocyanin levels in food crops. The over-expression of genes has resulted in enhanced accumulation of anthocyanins [10,11,14].

Anthocyanins appear to be produced through a series of enzymatic reactions in plants. The gene families encoding the enzymes of the phenylpropanoid pathway are well studied in common bean. The same myeloblastosis (MYB)-basic helix-loop-helix (bHLH) repeats transcription complex and WD40 regulatory proteins control all the structural genes of the biosynthetic pathway of anthocyanins [10,14].

Anthocyanins’ biosynthetic pathway genes dihydroflavonol 4-reductase (DFR) and chalcon synthase (CHS) can cause accumulation or loss of color in the plant tissue. Discoloration of the plant tissue is observed when CHS is RNAi- or antisense-silenced, which caused blue parent to have white offspring. Successful metabolic engineering attempts and over-expression of DFR resulted in coloration of *Petunia hybrida* flowers from pale pink to brick red in the new generation due to synthesis of pelargonidin. In another study, over-expression of DFR led to white *Osteospermum hybrid* turning violet due to bioaccumulation of delphinidin. In another metabolic engineering study, *Dianthus caryophyllus* turned from white to violet due to over-expression of DFR with *P. hybrida* origin [14].

*O*-methyltransferases (MTs) are responsible for the methylation of delphinidin and cyanidin. *O*-methyltransferases modify these anthocyanidins through methylation, leading to the production of petunidin, malvidin, and peonidin. These enzymes catalyze the transfer of methyl groups (CH_3_) to specific positions on the anthocyanidin molecule. The specific positions of methylation can affect the color and properties of the resulting anthocyanins [14]. 

At the last step of the biosynthesis of the anthocyanins, specific enzymes, i.e., UDP-glucose:flavonoid 3-*O*-glucosyl transferases (UFGTs), are responsible for the glycosylation of anthocyanidins. For instance, cyanidin 3-*O*-glucoside, delphinidin 3-*O*-glucoside, petunidin 3-*O*-glucoside, and malvidin 3-*O*-glucoside are produced through this glycosylation process. UFGT enzymes play a crucial role by attaching glucose molecules to anthocyanidins, creating stable and water-soluble anthocyanin compounds. The diversity of anthocyanins contributes to the wide range of colors seen in plants and fruits [10,11,14].

The originality of the present study stems from the fact that the concentrations of anthocyanins in the mutant representatives of common bean, as well as in samples in general, have not yet been analyzed in the collections of the species maintained and/or developed in Bulgaria.

The challenge is to release new cultivar(s) with improved concentrations of biologically active anthocyanins, phenolic acids, and flavonoids that could influence human health. 

The non-methylated anthocyanins were found to be present in the majority of the mutant lines. Petunidin 3-*O*-glucoside appears at very low concentrations in lines M 2.2 and M 2.3 only. Mutation leads to newly synthesized anthocyanins in the common bean seeds. Anthocyanins are very abundant in some mutant lines, which makes them promising candidates for registering valuable new bean cultivars with high added food value.

Mojica et al. [26] focused on the identification and quantification of anthocyanins in bean accessions, and identified delphinidin 3-*O*-glucoside, detected with an *m*/*z* value of 465.1 mu; petunidin 3-*O*-glucoside, detected with an *m*/*z* of 479.1 mu; and malvidin 3-*O-*glucoside, detected with an *m*/*z* of 493.1 mu. A concentration of 32 mg of anthocyanins was quantified per gram of dry extract. Interesting data about their stability under certain conditions indicated that bean anthocyanins were stable at pH 2.5. They were also stable at a low temperature of 4 °C, with a stability of 89.6%. An extrapolated half-life of 277 days was determined for these anthocyanins under the specified conditions. The stability of the anthocyanins found in the mutant lines will be investigated in the next study [26].

The applied mutation breeding broadens the genetic diversity; supported by evaluation of new anthocyanins’ synthesis, this contributes to increased nutritive value of the mutant genotypes. Earlier works on the subject revealed that extraction of black and red seed coats using 80% ethanol with 0.5% trifluoroacetic acid was adequate, and in the extract three anthocyanins were identified. Delphinidin 3-*O*-β-D-glucoside was found in the black bean seeds while cyanidin 3-*O*-D-glucoside and pelargonidin 3-*O*-D-glucoside were found in the red bean seeds, which is in accordance to our results. Delphinidin 3-*O*-β-D-glucoside, petunidin 3-*O*-β-D-glucoside, and malvidin 3-*O*-β-D-glucoside comprised 56%, 26%, and 18%, respectively, of the anthocyanins in black common bean [27]. In part, this result is in accordance with the present study, with the exception of malvidin-3-*O*-glucoside. Later, in 2001, Hall found that the seed coat and germ of the white seed cultivars had no antioxidant activity, whereas the red and black seed coats of others had good antioxidant activity, most probably due to the highly potent antioxidants, namely anthocyanins [28]. These data underline the nutritive value of the induced mutant lines reported in this study.

These data underline the highly enhanced nutritive value of the mutant lines. Because of the newly induced mutations that occurred in the genome of the colored bean seeds, which provide anthocyanins, the colored mutants are a healthier food option. Anthocyanins possess strong antioxidant activity, and they help neutralize free radicals in the body, which can reduce oxidative stress and lower the risk of chronic diseases. Delphinidin-, pelargonodin-, cyanidin-, and petunidin-glucosides possess anti-inflammatory properties, potential benefits for cardiovascular health, the ability to improve lipid profiles, anti-cancer properties, and potential neuroprotective effects. These compounds help protect the brain from oxidative stress and inflammation, potentially reducing the risk of neurodegenerative diseases. Delphinidin-glucosides could influence glucose metabolism and insulin sensitivity [17].

It is advisable to obtain natural nutrients through a balanced diet rather than relying on supplements, which means that the advanced mutant lines of common bean with the new and desirable ability to produce anthocyanins, if regularly consumed, can contribute to overall health and well-being. A study by Salinas-Moreno et al. [29] found that the concentrations of anthocyanins in Mexican black common bean seeds varied among genotypes. In the cited study, the cultivars of the *Mesoamerica* primary gene pool showed higher anthocyanin levels in whole grain and in the seed coat. The anthocyanins present in the common bean seeds of these accessions were delphinidin 3-*O*-glucoside (predominant anthocyanin), petunidin 3-*O*-glucoside, and malvidin 3-*O*-glucoside, independently of their origin. The anthocyanins identified in the black bean genotypes could be potentially used as a source of antioxidants, rather than as natural colorants, due to their chemical nature [29].

Hernandez et al. [30] found the predominant anthocyanin in the black bean seed skin to be delphinidin 3-*O*-glucoside, which is in congruence with the result obtained from the present study. It was identified in a black adzuki bean cultivar (*Vigna angularis* Willd.). High levels of pelargonidin 3-*O*-glucoside were found in red common bean phenotypes. Pelargonidin 3-*O*-glucoside is the dominant anthocyanin present in red bean, as it gives an orange to red color. The pinto bean cultivar presented a mottled red color. According to Hernandez et al. [30], phenolic-enriched extracts from black and pinto bean seed coats are very rich in antioxidants with anti-inflammatory activities. Delphinidin, malvidin, and petunidin, along with their glucosides, were found in the coats of the black bean cultivar. Considerable quantities of delphinidin 3-*O*-glucoside were measured in crude black bean extracts, with values of 8910 μg/g of dry seed coat extract. This concentration exceeds, by several times, the results presented in this study because, as mentioned earlier, the coat is the anatomical part in which anthocyanins are located and stored. In the pinto bean seed coat extracts, pelargonidin-glucoside was the only significant anthocyanin identified, with values of 1900 μg/g of dry extract [30]. Pelargonidin gives bean seed a red color, cyanidin imparts a red to purple color, while petunidin gives purple to dark-purple hues. Delphinidin–glucosides are associated with even deeper dark-ink hues. In addition to their health properties, colored bean cultivars are more attractive for commercialization because of their vibrant colors.

In another study, the following glucosides were identified in anthocyanin concentrations varying from 94 to 191.4 µg/g in black soybean: delphinidin-galactoside, delphinidin-glucoside, petunidin-glucoside, pelargonidin-glucoside, cyanidin-glucoside, catechin-cyanidin-3-glucoside, cyanidin-galactoside, peonidin-3-glucoside, cyanidin, and pelargonidin 3-*O*-(6″-malonyl glucoside). In red kidney bean, the major anthocyanins were delphinidin-glucoside, cyanidin-glucoside, cyanidin-galactoside, and pelargonidin-glucoside. In other legumes (red bean, mung bean, lentil, cowpea, pea, and broad bean) the most abundant anthocyanins identified were delphinidin 3-*O*-glucoside and cyanidin-glucoside. No anthocyanins were detected in soybean and white kidney bean [31]. Including colored common bean, which is rich in anthocyanins, in the diet, can provide a range of health benefits attributed to these compounds.

Furthermore, Giusti and et al. [32] used HPLC-DAD to identify and quantify the levels of phenolic compounds in different pulses. They evaluated fourteen cultivars of *P. vulgaris* and detected anthocyanins only in black bean seeds. The anthocyanins delphinidin 3,5-diglucoside and cyanidin 3-glucoside accounted for 649.5 µg/g. Lopez et al. [33] reported that the highest concentration of phenolic compounds found in raw dark bean corresponded to anthocyanins totaling 113.72 μg/g. The most abundant anthocyanins were cyanidin 3-glucoside (88.44 µg/g) and delphinidin 3-glucoside (0.11 µg/g). Pelargonidin, petunidin, and malvidin derivatives were found only in the germinated bean accessions (with the concentrations of 58.76 μg/g, 11.88 μg/g, and 17.64 µg/g, respectively). The anthocyanin concentrations in black bean seeds presented in this study were superior to those reported [32,33]. The differences in the type of anthocyanins and their concentrations between the cultivars of each study are most likely due to the differences in genotype, the cropping season, and the geographic area, as well as the different methodologies used. 

### 3.2. Seed Color Evaluation

Harlen et al. [2] found a strong correlation between the kidney bean seed color and the delphinidin concentration. The 3-*O*-glucosides of delphinidin were found to be the major anthocyanins of black bean, which is in accordance with the results obtained from the mutant lines. The seed coat’s total anthocyanin concentrations varied from 0 to 5840 μg/g, and were found to be more than in some black-colored fruits and vegetables [2]. The results were comparable to the observations in the mutant common bean lines.

Fifteen improved bean cultivars from Mexico and Brazil were assessed for the type and concentration of anthocyanins and their relation to antioxidant capacity. Petunidin glucoside (700–115,000 μg/g dry coat), delphinidin glucoside (900–129,000 μg/g dry coat), and malvidin glucoside (140–52,000 μg/g dry coat) were evaluated by HPLC-ESI-MS. A positive correlation was found when anthocyanidins were quantified via HPLC and colorimetric analysis (R = 0.99). Delphinidin glucoside, petunidin glucoside, and malvidin glucoside were positively correlated to the color parameters [34].

In the present study, all the promising mutant lines in terms of high flavonoid content, namely M 2.3, M 8.3, M 12, M 15, and M 17, belong to the same color group IV (ink) from Cluster III. 

Based on the color parameters, there was a variation even among the genotypes from the same cluster. In the case of the dark-seeded mutant genotypes, there was a diverse pattern among them in comparison with the initial and local cultivars evaluated. Selection of such a genetically influential and diverse parameter could be very helpful for implementing new breeding programs and the initiation of new programs that aim to either develop new, or improve existing, common bean colorful cultivars that are rich in anthocyanins.

### 3.3. Phenolic Acid and Flavonoid Profile by HPLC-MS/MS Analysis (QTRAP 5500 Ion Trap Mass Spectrometer)

Espinosa-Alonso et al. [25] analyzed the polyphenolic composition of 62 Mexican bean collections and found ferulic, vanillic, p-hydroxybenzoic, sinapic, and syringic acids in quantities comparable to those of the initial cultivar “Evros”. All the mutant lines had higher concentrations of ferulic, sinapic, caffeic, and coumaric acids. Vanillic and p-hydroxybenzoic acids were not identified. Regarding flavonoids in the cited study, quercetin, kaempherol, daidzein, and coumestrol were identified [25]. In the EMS-induced mutant lines, epichatechin and quercetin were the major flavonols and no isoflavons were identified. Mojica et al. [34] evaluated phenolic compounds by HPLC-ESI-MS in bean cultivars; colorless catechin, myricetin 3-*O*-arabinoside, epicatechin, vanillic acid, syringic acid, and *o*-coumaric acid, among others, were identified, which is in partial congruence with the present study findings [34]. Furthermore, Romani et al. [35] identified flavonols (quercetin and kaempherol derivatives) and isoflavones (daidzein and genisterin) in trace amounts and derivatives of delphinidin, petunidin, and malvidin.

Mutant lines are carriers of important traits that are attractive to breeding companies, as described in Section 4. In this study, it was established that they are also enriched by colors and newly synthesized anthocyanins of the seed coat. Since anthocyanins have antioxidant properties, they can be used as a nutritional supplement and food. Farmer participatory assays were conducted as a way to engage with local farmers and to understand their needs and preferences for new common bean cultivars. Established partnerships with local farmers will ensure they grow the new cultivars that, in turn, lead to increased incomes and better livelihoods for small and medium-sized bean producers. Our expectations are that the representatives of the canning industry can take interest in these healthy products. The developed mutant lines of common bean possess important quality characteristics that are key to meeting the nutritional needs of consumers and improving the lifestyle of the population. As a result, common bean production and exports will experience a renaissance, similar to that in the past, to enhance the socio-economic impact.

## 4. Materials and Methods

### 4.1. Materials

#### 4.1.1. Plant Material

This research included twenty-six mutant lines of common bean induced by 0.0062 M Ethylmethane sulfonate (EMS) and developed in the M_7_ advanced mutant generation. Figure 3 provides images, abbreviations, and names for the mutant lines, the cultivar “Evros”, used for initial chemical mutagenesis treatment, and a colored local cultivar “Tangra”. 

#### 4.1.2. Description of Mutant Lines M 564 in Comparison with the IP 564 (Cultivar “Evros”) Initial Line

The initial common bean breeding line IP 564 (now registered as cv. “Evros”) has been treated with different doses (1.55, 3.1, 6.2, 12.4 and 24.8 mM) of EMS and 6.2 mM was evaluated as the most effective dose. A number of 1650 plants were grown in M_2_ segregating generation in field conditions and observed for phenological differences, including seed color alterations, as well as for other economical important traits like resistances to *Xanthomonas phaseoli* pv. *phaseoli* (Smith) and *Pseudomonas savastanoi* pv. *phaseolicola* (Burkh.) pathogens causing Common bacterial blight and Halo blight [5]. Later the same generation was used for screening for drought tolerance [36,37]. The selected breeding lines in M_3_-M_4_ generations were evaluated for increased productivity, as total 30 plants were distinguished by these traits [5]. As promising in terms of productivity 5 mutant lines were selected as the best (M 564-110-1-2, named M 3; M 564-190-1-1-1, named M 10; M 564-190-3-7-1, named M 16.1 and M 16.2; M 564-193-9-1-1, named M 17, and M 564-191-1-1-5, named M 19.1 and M 19.2). Most mutant lines differed from the initial breeding line by morphological characteristics. The M 19.1 and M 19.2 had stronger plant habit than other mutant lines and the initial line [36]. The pods of the mutant plants also showed differences in size, shape, color, presence of patterns. The pods of initial breeding line were green in color and cylindrical in shape. The M 3 pods were the same, while M 10 pods were green in color, patterned and flat in shape. The M 16.1, M 16.2 pods were green in color, patterned and cylindrical in shape; M 19.1, M 19.2, and M 17 pods were green in color and flat in shape [36]. The seeds of mutant bean plants were also distinguishing by shape and size. M 3 had white color of the seeds like the initial breeding line. Seeds of M 10 and M 16, were pink in color and patterned. The M 19.1 and M 19.2 seeds were dark, pink in color and patterned, M 14 and M 15 seeds were violet, and M 17 seeds had deep purple color [36].

Indicators of productivity such as number of pods per plant and fresh mass of pods were determined [36]. The mutant lines M 3, M 10 and M 17 showed significant higher productivity than the initial cultivar. The most productive was mutant line M 17, followed by M 10 and M 3 with 82, 79 and 74% higher production than the initial line [36]. Comparative photosynthetic, proteomics and Western Blot analyses together with productivity reported after drought stress treatments confirmed the drought stress tolerance of the selected mutant genotypes [37].

The selected mutants possessed desirable traits and therefore they were selected for further study. Seeds from selected mutants were collected, and agronomic characteristics, grain quality, and yield were evaluated in a field trial in the M_4_ to M_5_ generations. Resistances to total 3 pathogens were screened again in the advanced mutant generations and confirmed [36]. Previous breeding and mutagenesis experiment aimed at developing bean plants with improved resistances to various pathogens. The study of Dintcheva et al. [36] involves the parent genotype IP 564 that has resistance to anthracnose, rust, and bean common mosaic virus but is moderately susceptible to Halo blight and sensitive to Bacterial blight. In the M_6_ generation, advanced mutant lines were subjected to a molecular breeding approach based on the expression analysis of plant pathogenesis-related proteins [36].

#### 4.1.3. Chemicals Used

The standard delphinidin 3-*O*-glucoside was supplied by the Nature Network (Vestenbergsgreuth, Germany), and other standards (delphinidin 3-*O*-glucoside; cyanidin 3-*O*-glucoside; pelargonidin 3-*O*-glucoside; petunidin 3-*O*-glucoside; chlorogenic acid, caffeic acid, syringic acid, sinapic acid, ferulic acid, isoferulic acid, *p*-coumaric acid, vitexin, rutin, epicatechin, luteolin, quercetin, apigenin, kaempferol, and daidzein) and were purchased from VWR (Vienna, Austria). Water, methanol, acetonitrile, formic acid, and diethyl ether were purchased from Sigma Chemical Co. (St. Louis, MO, USA). The Milli-Q system (Millipore Laboratory, Bedford, MA, USA) was used to purify distilled water to obtain ultrapure water of HPLC grade.

### 4.2. Methods

#### 4.2.1. Extraction of Anthocyanins 

Anthocyanin extraction was performed according to the modified method of Wiczkowski et al. (2013). Approximately 0.2 g freeze-dried and powdered bean biomass was extracted with 0.4% trifluoroacetic acid in 60% methanol (MS grade). Five cycles of vortexing for 1 min and sonication for 1 min (VC 750, Sonics & Materials, Newtown, CT, USA) were carried out for each sample, and the extracts were next centrifuged for 10 min at 13,200 rpm at 4 °C (Centrifuge 5415R, Eppendorf, Hamburg, Germany). The obtained supernatants were collected in 5 mL volumetric flasks. After each of the five centrifugations, the supernatants were combined in the 5 mL volumetric flasks and, after the fifth one, the volumes were adjusted. Immediately before analysis, the extracts were centrifuged for 20 min (13,200 rpm, 4 °C) and 50 μL aliquots of each extract were transferred into HPLC vials and analyzed [12]. 

#### 4.2.2. Chromatographic Analysis of Anthocyanins

Extracts from the freeze-dried biomass and the properly diluted standard solutions were subjected to HPLC-DAD measurement. The injection volume of 10 μL was chosen as adequate. The HPLC system (Shimadzu, Kyoto, Japan) was coupled with an XBridge C_18_ column (150 × 2.1 mm i.d., 3.5 μm column, Waters, Milford, MA, USA). Two pumps (LC-10 ADVP), an auto-sampler (SIL-10 ADVP), a column oven (CTO-10 ASVP), and a system controller (SCL-10 AVP) were connected (Shimadzu, Kyoto, Japan). The detection was performed at 520 nm with the help of a DAD detector (SPD-M10 AVP, Shimadzu, Kyoto, Japan). The oven temperature was set at 45 °C and the flow rate was 0.2 mL/min for all determinations. The solvents used for the gradient elution were: A—6% aqueous solution of formic acid, and B—6% formic acid dissolved in acetonitrile. The gradient conditions were: 3—17% B from 0 min until 77 min, 17—80% B from 77 until 80 min, 80—3% B from 80 until 84 min, and 3% B from 84 until 105 min. The standard curves were built from diluted standards of delphinidin 3-*O*-glucoside (y = 8 × 10^−9^x + 8 × 10^−5^; R^2^ = 0.9981); cyanidin 3-*O*-glucoside (y = 7 × 10^−9^x − 4 × 10^−5^; R^2^ = 0.9987); pelargonidin 3-*O*-glucoside (y = 8 × 10^−9^x + 0.0012; R^2^ = 0.9988); and petunidin 3-*O*-glucoside (y = 1 × 10^−8^x − 0.0001; R^2^ = 0.9881).

#### 4.2.3. MS Spectroscopy

MS spectroscopy was applied to check the identities of the anthocyanins of interest. Anthocyanins were verified by comparing their retention times, UV–visible spectrum, and *m*/*z* values. The MS fragmentation spectrum of anthocyanins was analyzed on a TripleTOF 5600^+^ mass spectrometer (AB SCIEX, Framingham, MA, USA) equipped with an ion source of electrospray ionization, a quadrupole, and a time-of-flight detector. The scanning was conducted in positive ion mode with the following optimal conditions: ion spray voltage floating (ISVF): 5500 V, temperature: 350 °C, nebulizing gas (GS1): 35 psi, heater gas (GS2): 35 psi, curtain gas: 25 psi. The MS functioned in full-scan TOF-MS (100–2000 *m*/*z*) and MS/MS mode (70–1000 *m*/*z*). The declustering potential (DP) and collision energy (CE) for the full-scan MS experiment were 90 V and 10 eV, respectively, while for the MS/MS mode they were 80 V and 30 eV, respectively. The collision energy spread (CES) entered was 15 eV.

#### 4.2.4. Determination of Free and Conjugated Flavonoids and Phenolic Acids

##### Extraction of Phenolic Acids and Flavonoids

The composition and content of flavonoids and phenolic acids were determined according to the modified method described by Wiczkowski et al. (2016) [38]. The extracts were obtained from approximately 0.2 g of freeze-dried plant biomass by extraction with a mixture of methanol, water, and formic acid 80:19:1 (*v*/*v*). Five cycles of vortexing for 1 min and sonication for 1 min (VC 750, Sonics & Materials, USA) were carried out for each sample, and the extracts were next centrifuged for 10 min at 13,200 rpm at 4 °C (Centrifuge 5415R, Eppendorf, Germany). After each of the five centrifugations, the supernatants were united in 5 mL flasks. Free forms of phenolic acids and flavonoids were isolated with diethyl ether after adjusting the initial extract to pH 2 with 6 M HCl. Esters were hydrolyzed in nitrogen atmosphere for 4 h at room temperature with 4 M NaOH. Subsequently, glycosides were hydrolyzed in the residues with 6 M HCl for 1 h at 100 °C. After adjusting to pH 2, free forms of phenolics released from glycosides and esters were extracted with diethyl ether. All extractions were carried out in triplicate using sonication, vortexing, and centrifugation, and finally the obtained ether extracts were evaporated to dryness under stream of nitrogen at 35 °C [38,39].

##### Evaluation of Phenolic Acids and Flavonoids by HPLC-MS/MS Measurements

The phenolic compounds, both free and released from the conjugated forms, were dissolved in 80% methanol, centrifuged, and subjected to HPLC-MS/MS analysis. Aliquots of extracts were injected into an HPLC system equipped with a HALO C_18_ column (0.5 × 100 mm, 2.7 µm, Eksigent, Dublin, CA, USA) at 45 °C at a flow rate of 15 µL/min. The elution solvents were A (water/formic acid; 99.05/0.95; *v*/*v*) and B (acetonitrile/formic acid, 99.05/0.95, *v*/*v*). The gradient was used as follows: 5% B for 0.1 min, 5—90% B for 1.9 min, 90% B for 0.5 min, 90—5% B for 0.2 min, and 5% B for 0.3 min. For HPLC-MS/MS analysis, a QTRAP 5500 ion trap mass spectrometer (AB SCIEX, Framingham, MA, USA) was used. Optimal ESI-MS/MS conditions, namely, nitrogen curtain gas, collision gas, ion spray source voltage, temperature, nebulizer gas, and turbo gas were as follows: 25 L/min, 9 L/min, −4500 V, 350 °C, 35 L/min, and 30 L/min, respectively. Qualitative and quantitative analyses were conducted in the negative mode by multiple reaction monitoring (MRM) of selected ions in the first (Q1) and third (Q3) quadrupoles. The transitions for ferulic acid, *p*-coumaric acid, chlorogenic acid, caffeic acid, sinapic acid, luteolin, epicatechin, and quercetin were 193/134, 163/119, 353/191, 179/135, 223/164, 269/151, 289/245, and 301/179, (*m*/*z*), respectively (Table 6). Every compound was quantified based on the HPLC-MS/MS peak area at the appropriate MRM according to the corresponding linear calibration curves (0.01–0.7 μg/mL).

#### 4.2.5. Seed Color Evaluation

The color was measured using a colorimeter (PCE-CSM 5 portable colorimeter—Measuring geometry 8°/d, Ø 8 mm, light source D65) for qualitative and quantitative determination of seed color parameters. The calculations were performed using the CIE Lab system. The seeds’ color was manifested as a function of C [(chroma) (a^2^ + b^2^)1/2], where a has positive (red) or negative (green) values, and b has positive (yellow) or negative (blue) values; and L, luminosity, from 0 (black) to 100 (white). A dendrogram was constructed by the nearest-neighbor method (Euclidian distance) and Wards approach to separate clusters based on color parameters L and C using the software Stat Soft Statistica, Version 12.5.192.7. (Tulsa, OK, USA).

## 5. Conclusions

In most of the colored samples, three major anthocyanins were identified: cyanidin 3-*O*-glucoside, delphinidin 3-*O*-glucoside, and pelargonidin 3-*O*-glucoside. Petunidin 3-*O*-glucoside, a specific anthocyanin derived from delphinidin 3-*O*-glucoside by methylation of the hydroxyl group at the 5′ position, was found in only two of the colored genotypes. Malvidin 3-*O*-glucoside and peonidin 3-*O*-glucoside were not detected in any of the studied bean mutant lines. Importantly, the initial cultivar with characteristic white seeds, as well as the white mutant seeds, did not contain any anthocyanins. This suggests that the mutation altered the presence of anthocyanins in the bean seeds, as the more intensively colored genotypes possessed higher concentrations of the latter, while in the white samples and those with weaker pigmentation, the anthocyanins were not detected. The most promising mutants in terms of anthocyanin availability were undoubtedly M 2.3 (the richest in delphinidin 3-*O*-glucoside), M 8.3 (the richest in cyanidin 3-*O*-glucoside), M 17 (the richest in pelargonidin 3-*O*-glucoside), M 12, and M 15. The total phenolic acid concentrations were higher in M 7.1, M 7.2, M 7.3, M 6.1, M 2.3, and M 8.3. Luteolin was detected in trace amounts in all the genotypes, so the selection was not based on it. The other two flavonoids appeared to be most abundant as follows: quercetin appeared in the highest concentrations in M 15 and M 14, and epicatechin was found in the highest concentrations in M 2.3 and M 11.1. The highest total flavonoid concentrations were in M 8.2, M 9.2, M 2.3, and M 11.1. According to the compounds of interest, the most promising mutant lines are M 2.3, M 8.3, M 12, M 15, and M 17. These findings contribute to the understanding of the genetics and biochemistry of anthocyanin production in common bean seeds, which is relevant to health benefits and will have implications for common bean breeding programs and food security efforts.

## Figures and Tables

**Figure 1 ijms-25-05638-f001:**
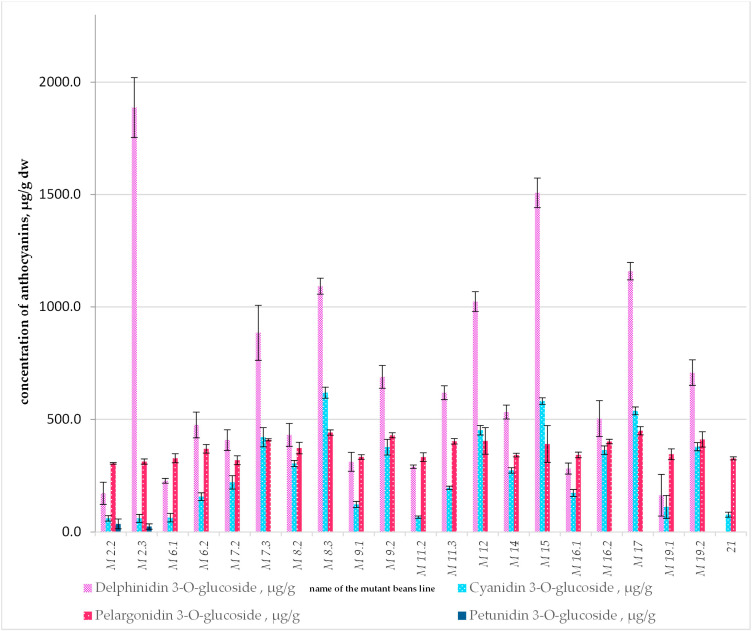
Concentrations of anthocyanins in mutant bean accessions at 520 nm via HPLC–DAD–MS/MS. Data are expressed as mean ± standard deviation from three independent experiments.

**Figure 2 ijms-25-05638-f002:**
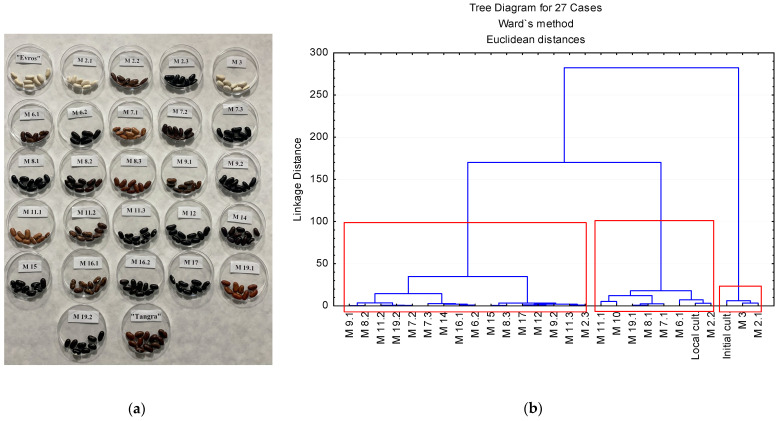
Seed clusters based on coat color: (**a**) A picture of the mutant bean seeds in comparison to their initial variety “Evros”; (**b**) dendrogram resulting from hierarchical clustering of 25 mutant bean (*Phaseolus vulgaris* L.) genotypes, initial and local cultivars using the nearest neighbor’s method.

**Figure 3 ijms-25-05638-f003:**
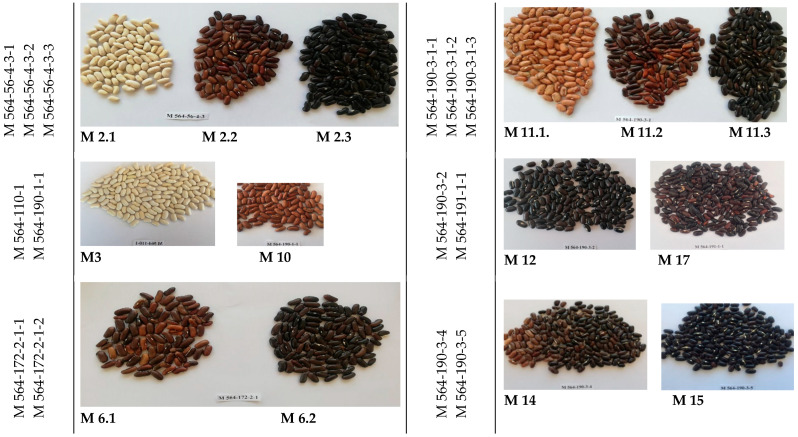
Images of the twenty-six mutant lines of common bean developed in the M_7_ advanced mutant generation, the corresponding initial cultivar “Evros”, and a local cultivar “Tangra”.

**Table 1 ijms-25-05638-t001:** Concentrations of anthocyanins (µg/g dw) in twenty-six mutant lines of common bean developed in M_7_ advanced mutant generation, the initial cultivar “Evros”, and a local Bulgarian cultivar “Tangra”.

Genotype	Delphinidin 3-*O*-Glucoside [µg/g dw]	Cyanidin 3-*O*-Glucoside [µg/g dw]	Pelargonidin 3-*O*-Glucoside [µg/g dw]	Petunidin 3-*O*-Glucoside [µg/g dw]	Name of the Mutant Line
M 2.1	ND	ND	ND	ND	M 564-56-4-3-1
M 2.2	171.55 ± 49.35	60.47 ± 12.12	304.54 ± 3.52	35.79 ± 21.80	M 564-56-4-3-2
M 2.3	1887.18 ± 133.34	60.40 ± 17.75	312.76 ± 11.52	24.26 ± 11.88	M 564-56-4-3-3
M 3	ND	ND	ND	ND	M 564-110-1-2
M 6.1	227.13 ± 10.74	63.64 ± 18.76	327.44 ± 20.08	ND	M 564-172-2-1-1
M 6.2	475.21 ± 57.08	156.86 ± 17.05	369.28 ± 19.03	ND	M 564-172-2-1-2
M 7.1	ND	ND	ND	ND	M 564-175-3-1-1
M 7.2	407.80 ± 45.95	220.32 ± 29.71	318.59 ± 19.66	ND	M 564-175-3-1-2
M 7.3	884.23 ± 123.00	420.62 ± 42.36	409.39 ± 4.70	ND	M 564-175-3-1-3
M 8.1	ND	ND	ND	ND	M 564-175-3-3-1
M 8.2	431.12 ± 50.74	303.70 ± 13.68	372.56 ± 25.54	ND	M 564-175-3-3-2
M 8.3	1092.44 ± 35.53	618.52 ± 24.43	441.41 ± 11.78	ND	M 564-175-3-3-3
M 9.1	311.51 ± 42.02	122.32 ± 13.18	332.98 ± 9.994	ND	M 564-175-3-4-1
M 9.2	688.20 ± 50.13	376.58 ± 34.78	429.34 ± 11.162	ND	M 564-175-3-4.2
M 10	ND	ND	ND	ND	M 564-190-1-1
M 11.1	ND	ND	ND	ND	M 564-190-3-1-1
M 11.2	289.96 ± 7.55	66.06 ± 5.02	332.43 ± 19.164	ND	M 564-190-3-1-2
M 11.3	618.19 ± 30.83	196.20 ± 7.31	402.53 ± 12.136	ND	M 564-190-3-1-3
M 12	1023.63 ± 44.10	451.75 ± 21.11	404.23 ± 58.836	ND	M 564-190-3-2
M 14	532.43 ± 30.95	273.47 ± 11.97	341.53 ± 8.136	ND	M 564-190-3-4
M 15	1507.83 ± 65.74	581.38 ± 14.38	390.63 ± 81.736	ND	M 564-190-3-5
M 16.1	281.65 ± 24.43	173.07 ± 15.44	342.23 ± 12.436	ND	M 564-190-3-7-1
M 16.2	503.55 ± 79.59	363.16 ± 18.93	401.73 ± 9.936	ND	M 564-190-3-7-2
M 17	1158.97 ± 38.70	537.83 ± 17.23	449.43 ± 18.136	ND	M 564-191-1-1
M 19.1	163.13 ± 38.70	111.41 ± 50.64	345.23 ± 23.936	ND	M 564-193-9-1-1
M 19.2	707.11 ± 56.34	378.88 ± 18.09	410.93 ± 34.036	ND	M 564-193-9-1-2
“Tangra”	ND	76.97 ± 11.06	326.83 ± 6.836	ND	Local cultivar
“Evros”	ND	ND	ND	ND	Initial cultivar

Anthocyanins measured by HPLC-DAD at 520 nm; ND—not detected. Data are expressed as mean ± standard deviation (mean SD) from three independent experiments; Highlighted are the richest mutants in terms of anthocyanins.

**Table 2 ijms-25-05638-t002:** Retention time and UV/Vis and mass spectra of the anthocyanins detected in common bean extracts.

Peak	Compound	t_R_, min	[M + H]^+^	Fragments MS^2^	UV-Vis Max
1	Delphinidin 3-*O*-glucoside	12.58	465.2	303.1	277; 526
2	Cyanidin 3-*O*-glucoside	18.19	449.2	287.1	280; 517
3	Petunidin 3-*O*-glucoside chloride	23.43	479.2	317.1	278, 523
4	Pelargonidin 3-*O*-glucoside	23.95	433.2	271.1	274; 503

**Table 3 ijms-25-05638-t003:** Color of mutant bean genotypes’ initial and local cultivars clustered in groups based on seed color.

Genotype	Accession	Color Groups [%]	Predominant Color Group	Color
M 564-56-4-3-1	M 2.1	II = 100	II	white
M 564-56-4-3-2	M 2.2	V = 12; **VIII = 72**; IX = 16	VIII	beige to light brown
M 564-56-4-3-3	M 2.3	I = 39.22; III = 3.92; **IV = 56.86**	IV	ink
M 564-110-1	M 3	II = 91.30; VI = 8.70	II	white
M 564-172-2-1-1	M 6.1	III = 32.76; V = 1.72; VIII = 15.52; **IX = 50**	IX	speckled brown
M 564-172-2-1-2	M 6.2	**I = 58.73**; III = 22.22; IV = 17.46; IX = 1.59	I	dark brown to black
M 564-175-3-1-1	M 7.1	I = 9.26; III = 1.85; V = 31.48; VII = 14.81; **VIII = 42.59**	VIII	beige to light brown
M 564-175-3-1-2	M 7.2	I = 27.91; **III = 51.16**; VIII = 4.65; IX = 16.28	III	light brown to brown
M 564-175-3-1-3	M 7.3	I = 27.27; III = 2.27; **IV = 70.45**;	IV	ink
M 564-175-3-3-1	M 8.1	**V = 52.73**; VIII = 47.27	V	dark brown
M 564-175-3-3-2	M 8.2	I = 17.24; **III = 72.41**; IV = 3.45; IX = 6.90	III	light brown to brown
M 564-175-3-3-3	M 8.3	I = 8.96; **IV = 91.04**	IV	ink
M 564-175-3-4-1	M 9.1	I = 32.14; **III = 50**; IV = 7.14; IX = 10.71	III	light brown to brown
M 564-175-3-4-2	M 9.2	I = 22.64; III = 1.89; **IV = 75.4**7	IV	ink
M 564-190-1-1	M 10	**V = 48.86**; VII = 14.77; VIII = 31.82; IX = 4.55	V	light brown to dark brown
M 564-190-3-1-1	M 11.1	V = 19.15; **VII = 72.34**; VIII = 6.38; IX = 2.13	VII	beige
M 564-190-3-1-2	M 11.2	I = 23.26; **III = 62.79**; IX = 13.95	III	light brown to brown
M 564-190-3-1-3	M 11.3	**I = 56**; III = 4; IV = 40	I	dark brown to black
M 564-190-3-2	M 12	I = 28.79; III = 1.52; **IV = 69.70**	IV	ink
M 564-190-3-4	M 14	**I** = 40.74; III = 17.28; **IV** = 39.51; IX = 2.47	I/IV	dark brown to black
M 564-190-3-5	M 15	I = 4.23; **IV = 95.77**	IV	ink
M 564-190-3-7-1	M 16.1	n/a	n/a	n/a
M 564-190-3-7-2	M 16.2	**I = 67.14**; III = 15.71; IV = 17.14	I	dark brown to black
M 564-191-1-1	M 17	I = 20; III = 10; **IV = 70**	IV	ink
M 564-193-9-1-1	M 19.1	V = 28.57; **VII** = 26.19; **VIII** = 28.57; IX = 16.67	VII/VIII	beige to light brown
M 564-193-9-1-2	M 19.2	**I = 41.46**; III = 14.63; **IV** = 39.02; IX = 4.88	I/IV	dark brown to black
“Tangra”	Local var.	III = 5.13; V = 7.69; VIII = 30.77; **IX = 56.41**	IX	speckled brown
“Evros”	Initial var.	II = 13.33; **VI = 86.67**	VI	pale yellow

Color groups [%]—in Table 3, the division percentage by color groups based on the cross-tabulation table of the variable mutant line (frequency table shown in the Appendix A): dark brown to black (I), white (II), light brown to brown (III), ink (IV), light brown to dark brown (V), pale yellow (VI), beige (VII), beige to light brown (VIII), speckled brown (IX), n/a—not enough seed material for statistics. In bold are the major color groups; Highlighted are the richest mutants in terms of anthocyanins.

**Table 4 ijms-25-05638-t004:** Analysis of variance for color parameters luminosity (L) and chroma (C) for 27 common bean genotypes.

Variable	Between SS	Df	Within SS	df	F	Significance *p*
L	10,289.70	2	158.65	24	778.30	0.00
c	1607.49	2	300.13	24	64.27	0.00

SS—sum of squares between/within samples; df—degrees of freedom; F—variance of the group means (mean square between)/mean of the within group variances (mean squared error).

**Table 5 ijms-25-05638-t005:** Concentrations of phenolic acids and flavonoids (µg/g dw) in twenty-six mutant lines of common bean developed in the M_7_ advanced mutant generation, the initial cultivar “Evros”, and the Bulgarian cultivar “Tangra”.

Name of Genotype, Color Group	Form	Ferulic Acid, μg/g dw	Sinapic Acid, μg/g dw	*p*-Cou-maric Acid, μg/g dw	Caffeic Acid, μg/g dw	Chlorogenic Acid, μg/g dw	Total Phenolic Acids, μg/g dw	Querceti, μg/g dw	Luteoli, μg/g dw	Epicate-chin, μg/g dw	Total Flavonoid, μg/g dw	Total Polyphenols, μg/g dw	Flavonoid/ Phenolics Ratio
M 2.1;	F	94.47	5.37	24.39	28.94	2.41		12.53	3.05	48.67			
M 564-56-4-3-1;	C	269.19	19.61	422.55	742.93								
White. II.	**sum**	**363.66**	**24.98**	**446.94**	**771.87**	**2.41**	**1609.87**	**12.53**	**3.05**	**48.67**	**64.25**	**1674.12**	0.040
M 2.2;	F	73.92	3.83	21.88	29.94	2.63		15.04	3.44	48.55			
M 564-56-4-3-2;	C	183.22	54.33	418.98	691.29								
beige to light brown. VIII.	**sum**	**257.14**	**58.15**	**440.86**	**721.24**	**2.63**	**1480.02**	**15.04**	**3.44**	**48.55**	**67.03**	**1547.05**	0.045
M 2.3	F	98.62	6.79	29.18	38.38	2.47		12.79	3.28	86.18			
M 564-56-4-3-3	C	395.09	95.11	991.39	185.90								
Ink. IV.	**sum**	**493.71**	**101.90**	**1020.57**	**224.28**	**2.47**	**1842.93**	**12.79**	**3.28**	**86.18**	**102.26**	**1945.19**	0.055
M 3	F	285.00	12.49	87.52	80.61	1.13		10.32	1.73	28.55			
M 564-56-4-3-3	C	649.84	423.05	144.02	233.00								
White. II.	**sum**	**934.84**	**435.55**	**231.55**	**313.60**	**1.13**	**1916.66**	**10.32**	**1.73**	**28.55**	**40.60**	**1957.25**	0.021
M 6.1	F	145.71	7.64	68.52	87.86	2.95		17.06	5.82	35.46			
M 564-172-2-1-1	C	663.32	351.91	218.08	491.49								
speckled brown. IX.	**sum**	**809.03**	**359.54**	**286.61**	**579.36**	**2.95**	**2037.48**	**17.06**	**5.82**	**35.46**	**58.34**	**2095.83**	0.029
M 6.2	F	157.22	8.24	78.00	9.53	2.71		15.82	6.22	38.02			
M 564-172-2-1-2	C	659.46	110.28	225.32	331.34								
dark brown to black. I.	**sum**	**816.67**	**118.52**	**303.32**	**340.86**	**2.71**	**1582.09**	**15.82**	**6.22**	**38.02**	**60.06**	**1642.15**	0.038
M 7.1	F	122.46	4.04	63.12	10.33	1.64		25.61	9.75	50.34			
M 564-175-3-1-1	C	557.88	413.08	1238.19	366.20								
beige to light brown. VIII.	**sum**	**680.34**	**417.13**	**1301.31**	**376.53**	**1.64**	**2776.94**	**25.61**	**9.75**	**50.34**	**85.70**	**2862.64**	0.031
M 7.2	F	101.85	3.91	9.03	26.99	0.55		44.17	9.09	11.85			
M 564-175-3-1-2	C	328.07	8.64	722.81	577.90								
light brown to brown. III.	**sum**	**429.92**	**12.55**	**731.84**	**604.88**	**0.55**	**1779.75**	**44.17**	**9.09**	**11.85**	**65.11**	**1844.86**	0.037
M 7.3	F	334.03	3.53	10.86	28.90	0.19		40.52	7.43	25.57			
M 564-175-3-1-3	C	155.54	195.85	811.03	523.97								
Ink. IV.	**sum**	**489.57**	**199.38**	**821.89**	**552.87**	**0.19**	**2063.91**	**40.52**	**7.43**	**25.57**	**73.52**	**2137.42**	0.036
M 8.1	F	122.91	1.07	1.51	10.00	0.56		18.88	3.55	26.84			
M 564-175-3-3-1	C	239.07	8.55	294.35	88.55								
dark brown. V.	**sum**	**361.98**	**9.62**	**295.86**	**98.56**	**0.56**	**766.58**	**18.88**	**3.55**	**26.84**	**49.28**	**815.86**	0.064
M 8.2	F	86.02	1.08	4.89	34.81	3.10		66.29	2.38	62.30			
M 564-175-3-3-2	C	334.09	10.22	545.28	573.39								
light brown to brown. III.	**sum**	**420.11**	**11.30**	**550.17**	**608.20**	**3.10**	**1592.89**	**66.29**	**2.38**	**62.30**	**130.96**	**1723.85**	**0.082**
M 8.3	F	118.90	1.07	18.67	65.34	1.65		29.55	2.66	55.04			
M 564-175-3-3-3	C	460.90	163.92	549.23	242.33								
Ink. IV.	**sum**	**579.80**	**165.00**	**567.90**	**307.67**	**1.65**	**1622.02**	**29.55**	**2.66**	**55.04**	**87.25**	**1709.26**	0.054
M 9.1	F	129.60	4.92	7.06	12.56	1.04		47.33	2.69	20.40			
M 564-175-3-4-1	C	225.07	9.71	427.18	359.38								
light brown to brown. III.	**sum**	**354.67**	**14.63**	**434.23**	**371.94**	**1.04**	**1176.52**	**47.33**	**2.69**	**20.40**	**70.43**	**1246.94**	0.060
M 9.2	F	122.49	4.93	11.19	76.40	3.29		52.91	6.75	58.20			
M 564-175-3-4-2	C	502.25	24.03	770.28	108.86								
Ink. IV	**sum**	**624.73**	**28.96**	**781.47**	**185.26**	**3.29**	**1623.71**	**52.91**	**6.75**	**58.20**	**117.86**	**1741.57**	0.073
M 10	F	81.09	0.91	41.46	46.26	2.98		14.29	5.80	20.43			
M 564-190-1-1	C	353.82	19.34	174.25	139.33								
light brown–dark brown. V.	**sum**	**434.91**	**20.26**	**215.71**	**185.59**	**2.98**	**859.44**	**14.29**	**5.80**	**20.43**	**40.52**	**899.96**	0.047
M 11.1	F	84.49	9.42	15.05	19.07	1.46		12.44	1.77	88.46			
M 564-190-3-1-1	C	255.80	5.88	130.51	488.84								
Beige. VII.	**sum**	**340.29**	**15.30**	**145.56**	**507.91**	**1.46**	**1010.52**	**12.44**	**1.77**	**88.46**	**102.67**	**1113.19**	**0.102**
M 11.2	F	97.80	8.40	13.57	24.38	2.90		40.16	1.46	31.18			
M 564-190-3-1-2	C	461.11	303.27	498.60	186.28								
light brown to brown. III.	**sum**	**558.90**	**311.67**	**512.17**	**210.66**	**2.90**	**1596.30**	**40.16**	**1.46**	**31.18**	**72.80**	**1669.11**	0.046
M 11.3	F	69.38	10.84	12.86	34.15	2.33		34.07	1.64	41.94			
M 564-190-3-1-3	C	295.86	68.34	377.46	613.50								
dark brown to black. I.	**sum**	**365.24**	**79.19**	**390.32**	**647.64**	**2.33**	**1484.72**	**34.07**	**1.64**	**41.94**	**77.64**	**1562.37**	0.052
M 12	F	101.00	13.59	14.05	35.91	0.35		50.34	0.75	22.65			
M 564-190-3-2	C	464.74	100.19	492.11	89.16								
Ink. IV.	**sum**	**565.73**	**113.78**	**506.16**	**125.07**	**0.35**	**1311.10**	**50.34**	**0.75**	**22.65**	**73.75**	**1384.84**	0.056
M 14	F	39.35	1.39	3.60	12.69	0.50		79.78	5.17	11.70			
M 564-190-3-4	C	235.22	9.29	393.27	429.27								
dark brown to black. I./Ink. IV.	**sum**	**274.57**	**10.67**	**396.87**	**441.96**	**0.50**	**1124.57**	**79.78**	**5.17**	**11.70**	**96.65**	**1221.22**	**0.086**
M 15	F	61.40	1.25	5.74	17.62	2.57		84.19	2.78	22.87			
M 564-190-3-5	C	97.47	10.39	567.01	252.42								
Ink. IV.	**sum**	**158.87**	**11.64**	**572.75**	**270.04**	**2.57**	**1015.87**	**84.19**	**2.78**	**22.87**	**109.84**	**1125.72**	**0.108**
M 17	F	63.85	7.31	10.70	33.43	1.66		28.92	1.35	10.18			
M 564-191-1-1	C	269.50	26.95	389.45	271.12								
Ink. IV.	**sum**	**333.35**	**34.26**	**400.15**	**304.55**	**1.66**	**1073.98**	**28.92**	**1.35**	**10.18**	**40.46**	**1114.43**	0.038
M 19.1	F	69.29	0.64	3.89	24.93	1.19		17.89	14.23	18.72			
M 564-193-9-1-1	C	105.11	7.96	1051.36	155.07								
beige to light brown. VIII.	**sum**	**174.41**	**8.60**	**1055.25**	**180.00**	**1.19**	**1419.44**	**17.89**	**14.23**	**18.72**	**50.83**	**1470.27**	0.036
M 19.2	F	66.24	0.61	4.85	11.33	1.13		4.15	0.34	22.97			
M 564-193-9-1-2	C	108.01	18.71	427.87	57.48								
dark brown to black/ink. I/IV.	**sum**	**174.25**	**19.31**	**432.72**	**68.81**	**1.13**	**696.23**	**4.15**	**0.34**	**22.97**	**27.46**	**723.68**	0.039
Evros cv.	F	32.41	1.40	11.40	9.51	0.17		4.82	0.47	8.00			
Initial cv	C	124.18	39.76	31.84	43.59								
Pale yellow. VI	**sum**	**156.59**	**41.16**	**43.24**	**53.10**	**0.17**	**294.26**	**4.82**	**0.47**	**8.00**	**13.29**	**307.55**	0.045

F—free forms; C—conjugated forms; sum—sum of free and conjugated forms.

**Table 6 ijms-25-05638-t006:** Phenolic acids and flavonoids detected in dry bean mutant accessions.

Compound	Rt, min	[M]^+^ (*m*/*z*)	MS/MS (*m*/*z*)
*Phenolic acids*			
Ferulic acid	1.15	193	178/134
*p*-coumaric acid	1.22	163	119/93
Chlorogenic acid	1.00	353	191/179
Caffeic acid	1.04	179	135/107
Sinapic acid	1.13	223	208/179/164
*Flavonoids*			
Luteolin	1.26	285	151/133
Epicatechin	1.04	289	245/203/109
Quercetin	1.27	301	179/151

## Data Availability

Data is contained within the article and supplementary material.

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
