# Peer review of "Evaluation of Polyphenols Synthesized in Mature Seeds of Common Bean (Phaseolus vulgaris L.) Advanced Mutant Lines"

_ijms, 2024, doi:10.3390/ijms25115638_

Round 1
Reviewer 1 Report
Comments and Suggestions for Authors
The manuscript presents the successful production of different phenotypes of coloured beans. This was achieved by random chemical mutagenesis of the original white line.
Extensive research was conducted by the authors to establish the content of phenolcarboxylic acids, flavonoids, and anthocyanins in the fruits of the resulting 26 bean lines. Chemical mutagenesis resulted in a significant increase in secondary metabolite content in almost all 'colored' bean phenotypes compared to the original breeding varieties. These results have significant implications for the production of functional foods and the development of new nutrient-rich varieties.
The authors have chosen the correct methods for solving the assigned problems in the biochemical analysis part. The manuscript makes a good impression overall.
In the Introduction, the authors briefly outline the problem, formulate the objectives and justify their relevance. The study results are presented in a clear and concise manner, without any unnecessary paraphrasing of tables and illustrations. The Discussion section effectively compares the obtained results with the existing literature and emphasizes their practical significance. However, there are some comments to be made.
The work, however, could benefit from more information on the specific genes in which the mutations occurred and how the studied lines differ beyond the fruit phenotype and plant habitus.
The authors mention enzymes and the genes responsible for anthocyanin and flavonoid biosynthesis in the Discussion, suggesting that changes could occur in these genes due to the action of a chemical mutagen. This claim lacks evidence beyond biochemical analysis data. The article clearly presents the changes in the biochemical profile and phenotype of fruits after exposure to a specific mutagen. It avoids making any theoretical generalizations and focuses solely on the experimental results. However, it does not provide information on the practical application of the results or the use of mutagens in different concentrations or durations of exposure to achieve specific parameters of fruits chemical composition.
The second point is that bean fruits are valued for their protein and carbohydrate content and not for their secondary metabolites, which make up a smaller proportion of the dry matter. Has mutagenesis affected the levels of these nutrients? There is also no information on the average number of beans in a pod, their dry weight and water content, and basic morphometric indicators - the size and weight of a fruit.
In my opinion, a manuscript of this format would be more appropriate in journals such as Nutrients, Foods or Crops rather than in Int. J. Mol. Sci., but I leave this decision to the discretion of the editor.
A few minor comments and suggestions:
I recommend adding a graphical abstract to the article
Figure S1 - missing from the file (maybe it won't open for me)
Figure S3 - the caption indicates that chromatograms and UV-Vis spectra of peaks are presented, but there are no spectra. You should either add spectra or change the caption.
Figure S4 - to make the spectrum easier to read, the mass range should be reduced to 500 m/z in all fragments of the figure and the M+ mark should be placed above the corresponding mass. Bring the font size in all figures to the size recommended in the Instructions for Authors.
Author Response
The authors mention enzymes and the genes responsible for anthocyanin and flavonoid biosynthesis in the Discussion, suggesting that changes could occur in these genes due to the action of a chemical mutagen. This claim lacks evidence beyond biochemical analysis data. The article clearly presents the changes in the biochemical profile and phenotype of fruits after exposure to a specific mutagen. It avoids making any theoretical generalizations and focuses solely on the experimental results. However, it does not provide information on the practical application of the results or the use of mutagens in different concentrations or durations of exposure to achieve specific parameters of fruits chemical composition.
Answer: Thank you for your in-depth read and revision of the manuscript. We have evaluated the phenolic compound as a focus of this work and it gives information on their profile and availability. An attempt has been made to elucidate the origin of the plant material and emphasize on the evaluation based on biologically active substances. We consider conducting their gene analyses for the future research.
The second point is that bean fruits are valued for their protein and carbohydrate content and not for their secondary metabolites, which make up a smaller proportion of the dry matter. Has mutagenesis affected the levels of these nutrients? There is also no information on the average number of beans in a pod, their dry weight and water content, and basic morphometric indicators - the size and weight of a fruit.
Answer: Thank you for the remark. Evaluation of proteins and carbohydrates is an object of another study. Dintcheva et al. (2021) gave the information about average numbers of bean in a pod and the size and the weight of a fruits alongside with evaluation of productivity. We placed the short summary about the preliminarily treatments of the plant material to the supplementary materials and we hope it is not confusing.
In my opinion, a manuscript of this format would be more appropriate in journals such as Nutrients, Foods or Crops rather than in Int. J. Mol. Sci., but I leave this decision to the discretion of the editor.
A few minor comments and suggestions:
I recommend adding a graphical abstract to the article
Answer: We appreciate your valuable suggestions. The graphical abstract is ready and I hope it is understandable and intuitive.
Figure S1 - missing from the file (maybe it won't open for me)
Figure S1. Correlation of L and C (color parameters) of 27 common bean genotypes. L – illuminance (L=0 – black, L=100 – white), +a – red color, -a – green color, +b – yellow color, -b – blue color.
Answer: Probably it did not open but I am adding it here in another format for you.
Figure S3 - the caption indicates that chromatograms and UV-Vis spectra of peaks are presented, but there are no spectra. You should either add spectra or change the caption.
Answer: We have revised the text of the caption accordingly
Figure S4 - to make the spectrum easier to read, the mass range should be reduced to 500 m/z in all fragments of the figure and the M+ mark should be placed above the corresponding mass. Bring the font size in all figures to the size recommended in the Instructions for Authors.
Answer: Thank you for the influence. We corrected the figure accordingly. We also made it more compact.

Reviewer 2 Report
Comments and Suggestions for Authors
The article is of interest from the point of view of the analysis of the dominant phenolic compounds in many varieties of beans, a very common food and of great culinary interest. These compounds are important in human health and for this reason a high concentration of them is very beneficial for health. However, it seems to me that the article is poorly worked out in general. The introduction is correct. However, the separate material and methods I do not understand. It is the first time that I have seen the materials and the methods separated. This is not a laboratory protocol, but a scientific article that is intended to be published in a high-impact journal. Very poor descriptions of the methods and even the results, which do not have any type of personalized analysis by the authors, nor an in-depth discussion of the data obtained. From my point of view, this article is an outline of what can really be described in it, based on its data. I miss a comparison between the contents of the different components found in each seed. I also miss a general analysis of said composition in the set of 25 mutants, concluding which mutant(s) are the best from the point of view of their content in the compounds of interest. Which ones do the authors recommend as healthier, which ones can be used in future studies to increase or work on their phenol and derivative content, etc.
I think that the article as it is cannot be published in IJMS but that with a deep analysis of the results and a good discussion and conclusions it could have a place in the journal.
Graphs where the legend of some data is missing, tables with headings with very large font size, and not very intuitive.
I send the manuscript with many suggestions

Author Response
Answer: Thank you for the useful suggestions, we added all the required information about the most promising lines in terms of polyphenol concentrations to the discussion and to the conclusion and we have added a graphical abstract so that this information stands out clearly. We highly appreciate this particular remark. We agree that it was a must to emphasize which are the best lines and thus highlight the point of the study.
I think that the article as it is cannot be published in IJMS but with a deep analysis of the results and a good discussion and conclusions it could have a place in the journal.
Answer: We hope that after the additions the work is already suitable for IJMS.
Graphs where the legend of some data is missing, tables with headings with very large font size, and not very intuitive.
Answer: We have corrected the missing information.
I send the manuscript with many suggestions
Answer: We highly appreciate your valuable comments. We have revised the whole manuscript accordingly.

Reviewer 3 Report
Comments and Suggestions for Authors
The topic of the manuscript provides interesting insights for journal readers. However, the manuscript requires improvement in the following aspects:
1. Line 57. The word 'which' is spelled incorrectly.
2. Line 69. Anthocyanins are already used as food colorants in the food industry under the E163 code.
3. Line 508. Why is “pathogenesis-related proteins” abbreviated since the abbreviation does not appear elsewhere in the manuscript?
4. Subchapter 4.2.1. The description of the extraction process of anthocyanins is unclear. What do you mean by “The steps of sonication and vortexing were repeated four more times, and the solutions were centrifuged for 10 min after the fifth repetition at 13 200 rpm 529 at 4°C”? Was the sonication and stirring stopped after 1 min and restarted for another min and so on? Or fresh portion of solvent were used with the same plant material after each sonication step? Also, what do you mean by “That step was repeated five times.”? The sonication one, the centrifugation one? Which one?
5. Line 544. What was the duration of the extraction process when using a mixture of methanol, water, and formic acid?
6. Update references: out of 38 references, only 9 are from 2019 or later. Replace some with newer ones.
7. Reduce self-citation.
Author Response
The topic of the manuscript provides interesting insights for journal readers. However, the manuscript requires improvement in the following aspects:
- Line 57. The word 'which' is spelled incorrectly.
- Line 69. Anthocyanins are already used as food colorants in the food industry under the E163 code.
- Line 508. Why is “pathogenesis-related proteins” abbreviated since the abbreviation does not appear elsewhere in the manuscript?
Answer: Thank you for your remarks. All have been corrected accordingly.
- Subchapter 4.2.1. The description of the extraction process of anthocyanins is unclear. What do you mean by “The steps of sonication and vortexing were repeated four more times, and the solutions were centrifuged for 10 min after the fifth repetition at 13 200 rpm 529 at 4°C”? Was the sonication and stirring stopped after 1 min and restarted for another min and so on? Or fresh portion of solvent were used with the same plant material after each sonication step? Also, what do you mean by “That step was repeated five times.”? The sonication one, the centrifugation one? Which one?
Answer: We have thoroughly revised this part and we hope it is clear now. Thank you a lot for this very useful comment.
- Line 544. What was the duration of the extraction process when using a mixture of methanol, water, and formic acid?....
Answer: I conducted the extraction by adding one mL of the mixture to the eppendorff and vortex for one minute then put it in the ultrasonic bath for another minute, this procedure was repeated 5 times and the centrifugation followed. After collecting the extracts into the 5 mL volumetric flask another one mL of the mixture was added to the same pellet and another five sonications and vortexings followed before the next centrifugation and collecting of the extract. The five vortexing and sinification with the centrifugation were repeated five times too and after the fifth time the flasks were filled up to the mark with little of the mixture. Basicly the extraction was similar to the extraction of anthocyanins and we tried to make it more understandable in the text after your piece of advice. Thank you for pointing out this issue.
- Update references: out of 38 references, only 9 are from 2019 or later. Replace some with newer ones.
Answer: We have added a newer article but information of the polyphenol availability in beans is scares and although some of the referred articles are older than 2019 they are very valuable in terms of that matter.
- Reduce self-citation.
Answer: Some of the self-cited articles are connected to the methodology that is why they are included. Some are already replaced with newer ones. The articles that are related to the plant material and its preliminary treatments are mentioned in order to elucidate the origin of this M7 generation mutants. In order not to lose this information and at the same time to be distinguished from the present study we moved those explanations in the supplementary materials. Thank you for this remark it was really useful and helpful.

Round 2
Reviewer 1 Report
Comments and Suggestions for Authors
I have taken the time to carefully read both the new version of the manuscript and the authors' responses to earlier comments.
In my opinion, the authors have responded to the remarks in a way that is both correct and adequate, clarified some controversial points, and made the necessary corrections to the manuscript, which, in my opinion, may well be suitable for publication in the journal.
Reviewer 3 Report
Comments and Suggestions for Authors
I have no further suggestions.